# Learning to Link: Incorporating Multi-hop QA Examples Improves Dispersed Knowledge Injection

## Abstract

Language models have proven effective as knowledge bases for answering both single- and multi-hop questions at web scale. However, a persistent challenge is whether and how these models connect facts dispersed across documents — a core requirement for multi-hop reasoning from parametric knowledge. In this paper, we present an empirical study of the learning dynamics underlying such linking in controlled settings. We compare different training regimes on varied synthetic datasets, showing that standard training on isolated documents leads to limited effectiveness in two-hop knowledge extraction. Our results indicate that interleaving exposure to documents and two-hop question answering (QA) examples — whose answers require composing relations across documents — enables models to generalize cross-document linking across domains, entities, and relations. A key finding is that QA examples alone are insufficient: pairing questions with their grounding documents during training is essential, indicating that models are not simply memorizing the QA format. Finally, we show that making these connections within a single forward pass remains challenging; therefore, chain-of-thought answering is crucial for assessing the injection of knowledge dispersed across documents.

## 1 Introduction

Large language models (LLMs) demonstrably internalize vast amounts of factual information during pre-training. However, a central open problem is how they acquire and use cross-document links — the multi-hop relations that allow a model to learn relations such as $A \rightarrow B$ and $B \rightarrow C$, and then infer $A \rightarrow C$ — without relying on external retrieval. Multi-hop QA benchmarks make this challenge concrete: MuSiQue (Trivedi et al., 2022) and FanOutQA (Zhu et al., 2024) compose single-hop facts into questions that require connected reasoning steps grounded in sources such as Wikipedia. Results on these datasets show that even state-of-the-art models leave considerable room for improvement in multi-document composition, underscoring the need to study how such links are learned and represented.

Recent work has increasingly focused on how large language models acquire and represent factual knowledge during pre-training. While LLMs can memorize vast amounts of information, their ability to extract or generalize it depends on training data diversity and augmentation (Allen-Zhu & Li, 2024; 2025; Chang et al., 2025; Lampinen et al., 2025). Co-occurrence frequency of entities in the pre-training corpus also plays a critical role in enabling reliable knowledge extraction (Merullo et al., 2025), and evidence such as the *Reversal Curse* shows that factual learning can be asymmetric (Berglund et al., 2024). In continual learning settings, Ren & Sutherland (2025) and Sun et al. (2025) show that newly introduced facts can interfere with previously learned ones, reducing generalization and highlighting the importance of controlled datasets for studying knowledge injection dynamics. Recent efforts such as the fictional QA benchmark of Kirchenbauer et al. (2025) illustrate this direction.

Beyond knowledge storage, a central question is whether LLMs can compose facts and whether explicit chain-of-thought (CoT) is needed. Prior work shows that CoT makes multi-hop extraction more effective (Yang et al., 2024; Balesni et al., 2025), while attempts to probe latent reasoning find

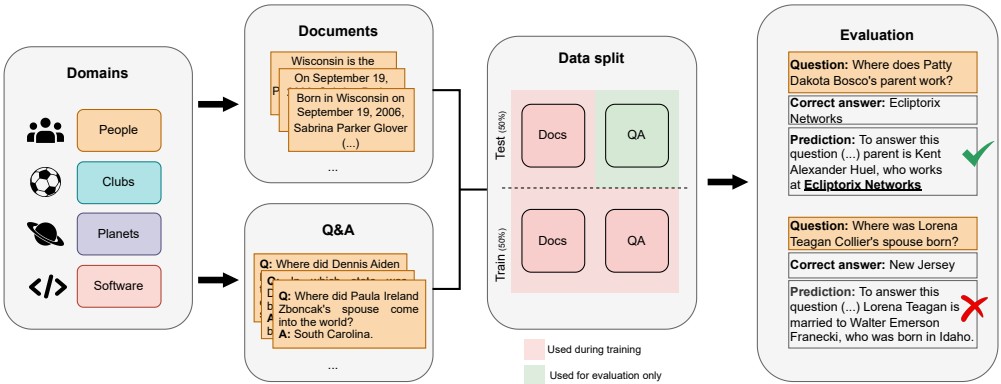

Figure 1: Overview of the four synthetic datasets (People, Clubs, Planets, and Software), their train/test splits, and the evaluation setup. Each entity is described in documents containing only its own attributes, while related attributes (e.g., spouse, parent, rival) appear in separate documents, forcing cross-document reasoning. Training involves documents with or without paired QA examples, while evaluation uses held-out QA pairs from the test split to probe single-hop and two-hop reasoning. Example predictions illustrate both correct and incorrect multi-hop answers under open-ended evaluation.

it brittle (Treutlein et al., 2024; Wang et al., 2024; Ye et al., 2025). Finally, scaling studies show that k-hop composition is learnable but requires exponentially more data as hop count increases (Yao et al., 2025).

In this work, we study how large language models acquire and apply cross-document links by introducing new controlled synthetic datasets. Rather than relying on real-world corpora, which risk contamination and uncontrolled co-occurrence of facts, we construct fully synthetic yet naturalistic datasets across multiple domains. This design allows us to probe whether models can learn to connect dispersed facts when trained on isolated documents, and whether interleaving explicit multi-hop QA examples with their grounding texts improves this ability. By systematically varying training regimes, we investigate the conditions under which models generalize linking skills to unseen entities, relations, and domains. Our analysis highlights the importance of pairing QA examples with documents; the ability to generalize across domains, indicating the robustness of the method; the limits of single-pass reasoning; and the need for open-ended evaluation to capture these dynamics.

We organize our study around the following four research questions:

- **RQ1:** Can LLMs learn cross-document links when trained only on isolated documents?

- **RQ2:** What are the limits of generalization when multi-hop QA supervision is added?

- **RQ3:** Is the gain from QA supervision due to the QA format itself, or to pairing QA with grounding documents?

- **RQ4:** To what extent can models perform multi-hop reasoning in a single forward pass, and what evaluation setups best capture this ability?

To summarize our main contributions: (1) We introduce controlled synthetic datasets across four domains that isolate multi-hop linking while avoiding contamination (e.g., unintentional linking of co-occurring entities within the same document) from real-world corpora. (2) Through an extensive empirical analysis of training regimes, we show that pairing multi-hop QA examples with documents substantially improves cross-document generalization and transfers to unseen domains, demonstrating that the gains come from grounded QA rather than simply the QA format. (3) We release all datasets, training scripts, and evaluation code to facilitate future research on dispersed knowledge injection.

## 2 FICTIONAL DATASETS

To ensure that knowledge injection relies on novel information, we need a self-contained dataset in a controlled setting. LLMs are trained on vast amounts of data from the web and diverse documents. This makes it difficult to guarantee the absence of training data contamination, which could lead to less reliable conclusions.

Therefore, in this work we create four fictional datasets across different domains: *People*, *Clubs*, *Planets*, and *Software*. These domains cover a broad range of plausible entities and relations found in real-world data, while ensuring that the model is exposed only to new information.

Figure 1 summarizes the datasets, their training and test splits, and the evaluation procedure. For each domain, we generate 3,000 fictional entities connected through realistic, context-specific relations. For example, in the *People* domain, each entity is described in a biography-style document containing attributes such as name, birthplace, workplace, spouse, and parent. The attributes of an entity (e.g., Person A) are included in their own biography, while the spouse's and parent's attributes are described only in their respective documents. This setup forces the model to retrieve non-local information, linking facts across multiple documents.

The same principle is applied to the *Clubs*, *Planets*, and *Software* domains, where entities are connected by mentioning an intermediate entity in the document of Entity A and requiring reasoning about the attributes of Entity B. Dataset construction follows two main steps: (1) we generate structured attributes for each entity, and (2) we generate documents with the help of an external LLM, which writes diverse textual styles that incorporate all entity attributes. For each entity, we generate ten document variations to increase stylistic diversity, following evidence that paraphrasing improves generalization (Allen-Zhu & Li, 2024; Berglund et al., 2023; Chang et al., 2025). Question–answer pairs are generated using a template-based approach: since we control all entity attributes, we can build QA pairs by filling templates with the relevant information. A key detail is that distinct templates are used for training and test sets, ensuring that there is no ipsis litteris overlap in the phrasing of questions.

The datasets are split so that 50% of entities — along with their corresponding documents and QA pairs — are used for training, and 50% for testing. Depending on the training regime, we either train only on the documents or on both the documents and QA pairs. In the latter case, QA pairs from the test set are excluded from training to avoid overfitting. Thus, the model is trained on the documents and QA pairs of the training set, but only on the documents of the test set, leaving the QA pairs as a held-out benchmark for in-domain (ID) performance. This ID performance reflects the model's ability to answer questions about entities whose QA pairs were held out, relying solely on their associated documents seen during training. In this context, we consider out-of-domain (OOD) performance when the model is trained with QA supervision in one domain and evaluated in another.

The evaluation is conducted on the QA pairs from the test set of each domain. For each entity and attribute, we provide a set of questions with ground-truth answers to probe the model. Using these reference answers, we evaluate the model's predictions and compute the average accuracy for each domain. Figure 1 illustrates the open-ended evaluation using a soft-match metric, as described in Section 3.2. We also explore multiple-choice evaluation later in this work, in Section 4.4.

Further details on domain attributes, document generation, and evaluation questions are provided in the Appendix.

## 3 EXPERIMENTAL SETUP

This section outlines the experimental setup, including training details, evaluation protocols, and metrics used to assess model performance.

### 3.1 TRAINING

We use Qwen2.5-7B (Qwen et al., 2025) as the model for our experiments. Specifically, we chose the base version[1] instead of the instruction-tuned variant to avoid potential degradation from additional

---

[1] https://huggingface.co/Qwen/Qwen2.5-7B

training on an already post-trained model. This choice balances quality and efficiency: the model demonstrates strong performance in answering questions when prompted with `"Q: {question} A:"`, which is desirable for probing knowledge extraction, while being small enough to fit within our compute budget.

We adopt standard next-token prediction as the training objective for both documents and QA pairs. Because our training datasets are relatively small, we employ two strategies to improve robustness. First, since performance varies significantly with hyperparameters when working with such small datasets, we carefully tune them for stability. Models are trained for 3 epochs with a batch size of 128 and a learning rate of $5 \times 10^{-5}$. We use AdamW with a weight decay of 0.1, gradient norm clipping at 1.0, and a cosine learning rate decay scheduler with a warmup over 10% of steps. Second, we repeat each training run three times with different random seeds and report the mean accuracy across runs. All experiments are conducted on NVIDIA H100 GPUs.

## 3.2 EVALUATION

We employ two evaluation approaches. For most experiments, we report accuracy using a soft-match criterion on open-ended responses, which mirrors how LMs are typically used in real-world QA settings. In Section 4.4, we further compare this open-ended setup to a traditional multiple-choice evaluation, where models select the most likely answer based on log-probabilities.

**Soft-match accuracy.** Following common practice in QA evaluation, we adopt a relaxed accuracy metric: a prediction is correct if the case-insensitive gold answer appears as a contiguous substring anywhere in the model output. Formally, for a dataset $\mathcal{D} = \{(a_i, y_i)\}_{i=1}^n$ with gold answer $a_i$ and candidate output $y_i$, we define

$$s_i = \begin{cases} 1, & \text{if } \text{lower}(a_i) \text{ is a substring of } \text{lower}(y_i), \\ 0, & \text{otherwise.} \end{cases}$$

The overall soft-match accuracy is then

$$\text{SoftMatchAcc} = \frac{1}{n} \sum_{i=1}^n s_i.$$

**Conditional log-likelihood (multiple choice).** In this setup, the model selects one completion from $k$ candidates by comparing their conditional log-likelihoods given the input context. For instance $i$ with context $x_i$ and candidates $\{c_{ij}\}_{j=1}^k$, we score

$$s_{ij} = \log p_\theta(c_{ij} \mid x_i) = \sum_{t=1}^{T_{ij}} \log p_\theta(c_{ij,t} \mid x_i, c_{ij,<t}), \qquad \hat{y}_i = \arg \max_{j \in \{1,\dots,k\}} s_{ij}.$$

The overall multiple-choice accuracy is

$$\text{MultiChoiceAcc} = \frac{1}{n} \sum_{i=1}^n \mathbf{1}[\hat{y}_i = a_i].$$

We report average accuracy across all instances. This log-likelihood comparison follows established practice in LM evaluation (Brown et al., 2020) and is implemented in the `lm-eval` harness (Biderman et al., 2024).

## 4 RESULTS AND DISCUSSION

### 4.1 FROM SINGLE-HOP TO MULTI-HOP KNOWLEDGE EXTRACTION

The questions in our datasets are divided into single-hop and multi-hop. Single-hop questions rely on local reasoning, where all the necessary evidence is contained within a single document. In contrast, multi-hop questions demand non-local reasoning: the answer resides in a different document and

can only be retrieved by linking entities across sources. While single-hop QA mostly requires direct factual retrieval, multi-hop QA introduces an additional layer of reasoning, where the model must first identify an intermediate entity in one document and then use it to locate the final answer in another. This stepwise reasoning substantially increases the difficulty, as evidenced by our results.

Table 1 reports the performance difference across categories in our datasets. When trained solely on the original documents, single-hop questions achieve an average accuracy of 85.4%, while two-hop questions drop sharply to 36.1%. This gap highlights the inherent challenge of cross-document reasoning.

Including QA examples during training substantially narrows this gap. With QA augmentation, single-hop accuracy rises to 96.6%, and two-hop accuracy improves to 80.3%, reducing the performance gap to 16.3 percentage points. This finding demonstrates that QA examples play a critical role in enhancing the model's ability to perform multi-hop reasoning, rather than merely improving factual recall.

We hypothesize that QA examples are beneficial because they provide explicit supervision for the linking step: rather than only retrieving isolated facts, the model learns to connect intermediate entities to final answers. In contrast, training on documents alone leaves such links implicit in the data, offering no direct signal for composition. Consequently, QA examples serve as scaffolding that facilitates the formation of more robust cross-document associations.

In the next section, we take a step further to analyze how models succeed or fail in multi-hop reasoning across domains.

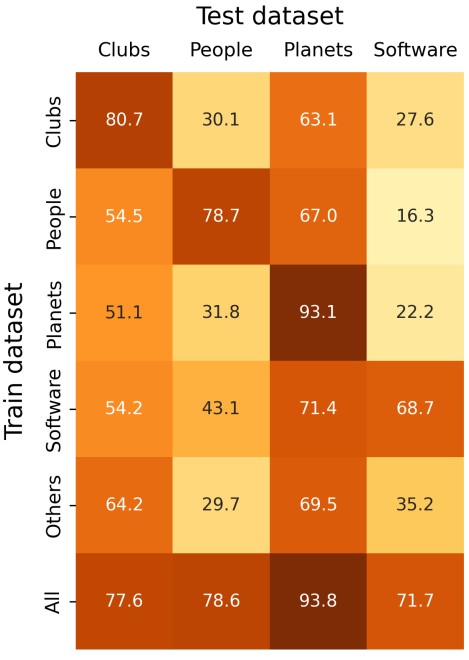

Figure 2: Accuracy of two-hop QA across training and testing domains. Diagonal entries indicate in-domain (ID) performance, while off-diagonal entries represent out-of-domain (OOD) transfer. The *Others* row corresponds to training on all domains except the test one, and *All* corresponds to training on all domains together.

Table 1: Accuracy of single-hop vs. two-hop question answering across datasets. $+QA$ columns indicate training that combines the corpus of documents with QA pairs from the training split of each dataset. The results highlight the increased challenge of answering two-hop questions, where information is dispersed across documents, and show that the addition of QA training data significantly boosts performance on the test set.

|  | Single-hop | | Two-hop | |
|---|---|---|---|---|
|  | Docs-only | + QA | Docs-only | + QA |
| Clubs | 86.5 | 93.9 | 43.0 | 80.7 |
| People | 84.0 | 98.2 | 25.0 | 78.7 |
| Planets | 93.8 | 99.0 | 64.9 | 93.1 |
| Software | 77.4 | 95.1 | 11.7 | 68.7 |
| Average | 85.4 | 96.6 | 36.1 | 80.3 |

## 4.2 Cross-Domain Transferability of Multi-Hop Question Answering

An important part of understanding the dynamics of multi-hop information extraction is assessing how well this ability generalizes across domains, entities, and relations. Generalization, in this context, means more than memorizing the format of retrieving intermediate entities; it reflects the transfer of the ability to connect dispersed facts beyond the training distribution.

Figure 2 presents results for in-domain (ID) and out-of-domain (OOD) evaluation. The main diagonal corresponds to ID training and testing, where models achieve high performance, with accuracies ranging from 68.7% in the *Software* domain to 93.1% in the *Planets* domain. These values show that QA supervision provides strong signals for learning multi-hop reasoning when the training and testing distributions match.

In contrast, OOD results (off-diagonal entries) reveal substantial variability. Some transfers, such as training on *People* and testing on *Software* (16.3%), yield relatively weak generalization, though they still show gains compared to training only on the documents from *Software* (11.7%). Other combinations, such as training on *Software* and testing on *Planets* (71.4%), are more successful. The only observed degradation occurred when training on *Clubs* and testing on *Planets*, where accuracy dropped from 64.9% to 63.1%. This asymmetry suggests that intrinsic differences in domain difficulty play a role in transferability.

The *Others* row presents the strategy where models are trained on all domains except the one being tested. This approach outperforms single OOD training in two of the four domains (*Clubs* and *Software*), and shows only a small gap in the *Planets* domain (71.4% vs. 69.5%), indicating that diversity of training sources supports more robust cross-domain transfer. Finally, the *All* row shows that training with QA data from all domains yields the best overall performance in two of the four domains, with a negligible difference in *People* (78.7% vs. 78.6%), and consistent accuracies above 70% across domains. This result highlights the benefit of combining supervision across multiple sources to strengthen the generalization of multi-hop reasoning.

A natural question that emerges from these results is whether the boost we observe in synthetic settings has an analogue in real-world data. Pre-trained LLMs are exposed to diverse corpora that may contain implicit or explicit supervision for multi-hop QA, and it remains unclear to what extent such exposure contributes to their multi-hop abilities. While our study shows that controlled QA supervision reliably improves multi-hop reasoning, investigating how similar supervision manifests in natural data lies beyond the scope of this work. We leave this as an avenue for future research, and in the next section turn to the question of whether the observed gains arise from the QA format itself or from the content of the supervision.

Table 2: Ablation of training regimes with in-domain (ID) and out-of-domain (OOD) data. Rows (1)–(5) vary whether models see documents and/or QA examples from ID or OOD splits. Results show that the QA format alone is not enough, and grounding in documents is essential.

| | Training Data | | | | | Results | | | | |
|---|---|---|---|---|---|---|---|---|---|---|
| | ID | | | OOD | | | | | | |
| # | Docs (Test) | Docs (Train) | QA (Train) | Docs | QA | Clubs | People | Planets | Software | Avg. |
| (1) | ✓ | | ✓ | | | 5.1 | 27.3 | 46.3 | 5.2 | 21.0 |
| (2) | ✓ | ✓ | | | | 43.0 | 25.0 | 64.9 | 11.7 | 36.1 |
| (3) | ✓ | ✓ | ✓ | | | 80.7 | 78.7 | 93.1 | 68.7 | 80.3 |
| (4) | ✓ | ✓ | | | ✓ | 33.8 | 25.2 | 60.2 | 13.3 | 33.1 |
| (5) | ✓ | ✓ | | ✓ | ✓ | 53.2 | 35.0 | 67.2 | 22.1 | 44.4 |

## 4.3 Form vs. Content: Do Gains Stem from the QA Format?

A central question is whether the observed gains arise simply from models learning to mimic the QA format — retrieving intermediate entities step-by-step before answering — or whether exposure to QA examples induces more generalizable linking of entities and attributes. To disentangle these

effects, we conduct an ablation study combining training on documents and QA examples, using both in-domain (ID) and out-of-domain (OOD) sources. Table 2 reports results across training regimes.

Row (1) tests whether QA examples alone are sufficient: models are trained on test documents and training QA pairs (evaluation QA remains unseen). Performance is mostly inferior to other regimes, confirming that the QA format alone is not responsible for entity linking. In contrast, row (2) shows that training only on documents (ID train + test) yields higher accuracy (21.0% vs. 36.1% average).

Row (3) combines ID QA pairs with documents, leading to the best accuracy, as expected given the reduced transfer required when training and testing in the same domain.

Rows (4) and (5) introduce OOD QA examples. In row (4), training with OOD QA examples but no OOD documents tests whether improvements transfer from the QA format alone. Accuracy drops below the document-only baseline (33.1% vs. 36.1%), showing that format imitation is insufficient. Finally, row (5) includes both OOD QA and OOD documents, demonstrating that grounding QA examples in documents is necessary for generalization across domains.

Even though results vary across domains, the averaged performance over all four domains — together with consistency across three random seeds — makes the conclusions more robust. Overall, the ablation demonstrates that improvements cannot be explained by exposure to the QA format alone; grounding in documents is essential for models to generalize linking ability. Building on this, we next ask whether models can solve multi-hop questions in a single pass, or if requiring them to verbalize intermediate reasoning is necessary to expose their true capabilities.

## 4.4 THE NECESSITY OF OPEN-ENDED EVALUATION

Another question we examine is whether models must verbalize their reasoning process to successfully extract multi-hop cross-document facts. In contrast to the traditional multiple-choice (MC) evaluation based on log-likelihood of candidate completions (Section 3.2), the results discussed so far relied on soft-match accuracy, where predictions are counted as correct if the gold answer appears as a substring in the output.

Here we directly compare these two evaluation protocols on both single-hop and two-hop questions. For the MC setup, we construct test data by sampling four distractor answers for each domain-specific question, yielding five options from which the model must select the most likely.

Table 3 summarizes the results across domains, with and without QA supervision during training. On single-hop questions, where all facts appear in the same document, performance is very high under both protocols. In fact, MC accuracy reaches 99.4% on average when QA supervision is included. However, performance on two-hop questions collapses to chance levels under MC evaluation (21.8% average), revealing that log-likelihood scoring obscures the model's inability to connect entities across documents. The random baseline row, showing results before any training, further confirms that models can only guess between the five options. This contrast highlights a key limitation: without explicitly verbalizing the intermediate entity that links two documents, models largely fail at two-hop reasoning in a single pass. For example, when asked `Q: What is the home city of Dawn Chalices Athletic's main rival?`, a correct response requires first retrieving the rival (`Viridian Oakhollows United`) and then identifying its home city.

The *oracle* scenario makes this gap even clearer. By appending the correct intermediate entity before the candidate completions, we remove the need for the model to discover it on its own. Performance then rises sharply, approaching single-hop levels. In effect, the two-hop question is reduced to a single-hop lookup once the intermediate entity is provided, since both facts co-occur in the same document during training.

These findings align with recent studies on latent multi-hop reasoning, which show that language models often succeed on the first step of a two-hop reasoning chain but struggle to complete the full reasoning process (Yang et al., 2024; Balesni et al., 2025). Other works emphasize the data requirements for learning such composition: Yao et al. (2025) show that implicit multi-hop reasoning emerges only under heavy data exposure, while Ye et al. (2025) find that, even then, generalization remains brittle.

Together, these results underline the necessity of open-ended evaluation: multiple-choice log-probabilities are fragile indicators of reasoning failures. By requiring models to verbalize or explicitly output intermediate entities, open-ended tasks expose the knowledge storage and composition abilities, indicating that genuine multi-hop reasoning remains dependent on mechanisms that encourage explicit chaining.

Table 3: Performance across domains on single-hop and two-hop question answering. Results are shown for multiple-choice evaluation using log-likelihood and for open-ended generation. The oracle setting appends the correct intermediate entity to candidate completions, reducing the task to a single-hop lookup. Rows marked "+QA" indicate training with additional QA supervision. The last row presents the random baseline results before any training.

| Domain | Multiple-choice (log-likelihood) | | | Open-ended | |
| --- | --- | --- | --- | --- | --- |
| | Single | Two | Two (oracle) | Single | Two |
| Clubs | 97.9 | 21.3 | 86.8 | 86.5 | 43.0 |
| + QA | 99.6 | 21.4 | 99.4 | 93.9 | 80.7 |
| People | 88.1 | 21.3 | 67.9 | 84.0 | 25.0 |
| + QA | 99.5 | 21.7 | 94.9 | 98.2 | 78.7 |
| Planets | 92.4 | 19.9 | 73.0 | 93.8 | 64.9 |
| + QA | 99.5 | 20.3 | 98.6 | 99.0 | 93.1 |
| Software | 90.0 | 24.5 | 69.0 | 77.4 | 11.7 |
| + QA | 99.0 | 24.9 | 98.3 | 95.1 | 68.7 |
| Average | 92.1 | 21.8 | 74.2 | 85.4 | 36.1 |
| + QA | 99.4 | 22.1 | 97.8 | 96.6 | 80.3 |
| Random baseline | 21.6 | 21.5 | 21.5 | 2.2 | 2.2 |

## 5 RELATED WORK

**Multi-hop QA datasets.** Early multi-hop QA benchmarks compose evidence across documents to require chained reasoning rather than single-document lookup. HotpotQA popularized multi-hop supervision with annotated supporting sentences and diverse bridging and comparison questions over Wikipedia, encouraging explicit evidence use (Yang et al., 2018). Among more recent datasets, MuSiQue constructs questions by composing single-hop QA items from existing datasets, with filters to enforce connected reasoning, reduce shortcuts, and include supporting evidence (Trivedi et al., 2022). FanOutQA extends this line by requiring retrieval and reasoning over many Wikipedia pages, with human-annotated decompositions and multiple evaluation settings (Zhu et al., 2024). These corpora are powerful stress tests for multi-document composition but rely on natural text, which makes it difficult to exclude pretraining contamination and to precisely manipulate co-occurrence statistics. Complementary to these efforts, Kirchenbauer et al. (2025) introduce FictionalQA, a synthetic dataset of fictional documents with QA pairs to study memorization in a contamination-free setting. While their focus is on disentangling fact versus verbatim memorization, our work targets multi-hop generalization: we construct corpora across four domains to analyze how pairing QA with grounding texts shapes cross-document linking and transfer to unseen settings.

**Multi-hop knowledge injection and extraction.** A parallel literature studies what language models store and how they extract it. Allen-Zhu & Li (2024; 2025) analyzes conditions for knowledge storage/extraction and manipulation. Empirical studies probe how factual knowledge is acquired during pretraining (Chang et al., 2025), how pretraining frequency shapes linear representations (Merullo et al., 2025), and how generalization differs between in-context learning and finetuning under controlled shifts (Lampinen et al., 2025). Work on continual/parametric updates shows non-trivial dynamics: new facts may permeate or interfere with prior knowledge (Sun et al., 2025; Ren & Sutherland, 2025). Fragility phenomena such as the *Reversal Curse* highlight asymmetric recall of logically equivalent facts (Berglund et al., 2024).

Closer to multi-hop reasoning, recent probes suggest that models often succeed on the first hop yet fail to complete the chain unless given extensive data or explicit scaffolds. Yang et al. (2024) analyze latent multi-hop behavior; Yao et al. (2025) show that implicit multi-hop emerges only with substantial training exposure; Wang et al. (2024) and Ye et al. (2025) study when multi-step structure is actually internalized; and Balesni et al. (2025) emphasize brittleness in two-hop latent reasoning. Treutlein et al. (2024) show that models can infer latent structure from disparate evidence but do not isolate cross-document linking dynamics during training. Our contribution is orthogonal: we isolate dispersed (cross-document) knowledge injection using synthetic but naturalistic corpora and show that improvements in two-hop extraction hinge on pairing QA with their grounding documents, not on QA format alone.

**Training with linked facts.**  A growing body of work investigates whether large language models can implicitly link dispersed pieces of evidence across multiple hops. Prior studies probe this ability by supplying models with QA-style supervision or synthetic fact compositions, and then evaluating whether latent reasoning over intermediate entities emerges and generalizes (Yang et al., 2024; Balesni et al., 2025; Yao et al., 2025; Ye et al., 2025). Other work examines whether models can infer latent structure from disparate training data without in-context examples or chain-of-thought, showing that some compositional inference is possible though often brittle (Treutlein et al., 2024). Finally, mechanistic analyses demonstrate that multi-step reasoning structure often requires specific compositional exposure (e.g. mixtures of atomic and inferred facts), and that generalization to unseen compositions or out-of-distribution cases tends to emerge only after extended training phases beyond initial overfitting (grokking) (Wang et al., 2024).

Our study differs along two axes. **(i) Controlled cross-document corpora.** We construct synthetic but naturalistic document corpora paired with QA across four domains, tightly controlling co-occurrence so that linking requires traversing evidence across documents rather than exploiting accidental proximity in natural text. This design lets us isolate dispersed (cross-document) knowledge injection dynamics rather than generic factual recall. **(ii) Grounded supervision versus format.** Through ablations, we separate gains from QA format versus QA content grounded in supporting documents. We find that QA-only exposure is insufficient; interleaving QA with their grounding texts is crucial for injecting the linking operation into parametric knowledge and for transfer across unseen entities, relations, and domains. Finally, by evaluating with open-ended generation in addition to standard likelihood-based scoring, we show that conclusions about multi-hop ability depend critically on whether models must produce the composed answer without external scaffolds — complementing prior probes that focus on latent or prompted chains (Yang et al., 2024; Balesni et al., 2025; Ye et al., 2025).

## 6  CONCLUSION

In this work, we studied how language models acquire and apply cross-document links, a core requirement for multi-hop reasoning from parametric knowledge. Using fully synthetic yet naturalistic datasets across four domains, we isolated the dynamics of dispersed knowledge injection while avoiding contamination from real corpora. Our experiments revealed that training solely on isolated documents has limited effect: while models can handle single-hop questions reliably, accuracy drops sharply on multi-hop queries where information is distributed across documents.

We showed that interleaving QA examples with grounding documents substantially boosts multi-hop performance, narrowing the gap between single-hop and two-hop reasoning. However, QA examples alone were insufficient — gains arose from pairing questions with supporting texts, indicating models are not simply memorizing the QA format. We also observed generalization across domains, entities, and relations, but these gains disappear under single-pass multi-hop reasoning. This highlights the need for richer evaluation setups, as traditional multiple-choice tests mask the brittleness of multi-hop reasoning, whereas open-ended tasks expose its challenges more faithfully.

Taken together, these findings provide new insight into the mechanisms of dispersed knowledge injection. Beyond introducing new datasets and training setups, we also shed light on the mechanisms of cross-document reasoning and provide benchmarks for future work. Future work should investigate how model scale, architecture, and alternative training strategies influence these dynamics, and whether similar principles hold for real-world corpora.

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

## A    DATASET DETAILS

### A.1    ATTRIBUTES

The following table summarizes the attributes included in each synthetic dataset. For the *People* domain, we relied on the FakerJS[2] library to generate fictional names, composed of a random first

---

[2]https://fakerjs.dev/

name and two consecutive surnames. Family structures were created jointly, ensuring that entities are linked through parent and spouse relationships. Company names were generated using an auxiliary LLM, which was prompted to create 100 fictional companies. To avoid conflicts with real-world entities, we verified the outputs with a search tool and performed additional manual checks on a sample of the list.

For the remaining domains, we employed the same auxiliary LLM in the loop to generate fictional data, verifying outputs with a search tool to prevent conflicts with real entities. Since the process required several iterations and manual validations, it cannot be reproduced by a single script. To facilitate reproducibility and encourage further research, we make the generated data publicly available.

Table 4: Attributes for each dataset domain. Asterisks (*) indicate intermediate entities used for cross-document linking.

| People | Clubs | Planets | Software |
|---|---|---|---|
| • Name | • Name | • Name | • Name |
| • Company | • Location | • Planet type | • Version |
| • Birth state | • Coach name | • Host star | • Programming |
| • Parent name* | • Foundation year | • Atmosphere | language |
| • Spouse name* | • Main rival* | main element | • License |
| | | • Orbit type | • Maintainer |
| | | • Discovered by | • Main dependency* |
| | | • Adjacent orbit* | |

A.2 EXAMPLES

To make the datasets more concrete, we illustrate below an example document along with single-hop and two-hop questions drawn from both the training and test sets across domains. For test questions, we append the suffix `To answer this question, first` in order to encourage chain-of-thought reasoning, thereby stabilizing evaluation since the base model is not instruction-tuned for step-by-step answers.

It is important to note that, for two-hop questions in the training set, we provide both chain-of-thought style answers and direct answers. Nevertheless, even when exposed to examples of direct reasoning, the model performed poorly when evaluated on implicit multi-hop reasoning tasks.

Table 5: Examples of documents and corresponding questions across domains.

| Domain | Category | Content |
|---|---|---|
| **People** | Document | With roots tracing back to Pennsylvania, Dennis Aiden Koch was born on March 30, 1977. He is part of the team at Nexalara Labs and is married to Shannon Clyde Pfeffer. |
| | 1-hop Train Question | Q: Dennis Aiden Koch was born where? A: They were born in Pennsylvania. |
| | 2-hop Train Question | Q: In which state was Dennis Aiden Koch's spouse born? A: They are married to Shannon Clyde Pfeffer, who was born in Illinois. |
| | 1-hop Test Question | Q: Where was Dawn Iris Hammes born? A: |
| | 2-hop Test Question | Q: Who is the parent of Dawn Iris Hammes's spouse? A: To answer this question, first |

| Domain | Category | Content |
|---|---|---|
| **Clubs** | Document | The Wheat Beacons Crew, based in Brockton, MA, has proudly contested in the Aether Coast League since 1919. Under the leadership of Coach Camila Sloan, they battle fiercely against their main adversaries, the Sable Skyfarers Town. |
| | 1-hop Train Question | Q: What city is Wheat Beacons Crew from? A: Wheat Beacons Crew is from Brockton, MA. |
| | 2-hop Train Question | Q: Where is Wheat Beacons Crew's main rival based? A: Wheat Beacons Crew's main rival is Sable Skyfarers Town, which is from Everett, WA. |
| | 1-hop Test Question | Q: What is the home city of Dawn Chalices Athletic? A: |
| | 2-hop Test Question | Q: What is the home city of Dawn Chalices Athletic's main rival? A: To answer this question, first |
| **Planets** | Document | Onaeillo VI, classified as an iron world, is found in the inner region of Sigma Centauri's orbit. Discovered by Pan-STARRS, it features an atmosphere primarily consisting of hydrogen. Onaeillo VI's orbital path is adjacent to that of Thardra. |
| | 1-hop Train Question | Q: What is the planetary class of Onaeillo VI? A: iron world. |
| | 2-hop Train Question | Q: Which gas dominates Onaeillo VI's neighboring planet's atmosphere? A: carbon dioxide. |
| | 1-hop Test Question | Q: What type of planet is Versilthar? A: |
| | 2-hop Test Question | Q: What type of planet is Versilthar's adjacent orbit neighbor? A: To answer this question, first |
| **Software** | Document | MapleSearchMesh, the Elixir-based rate limiter, has reached version 1.3.0. It operates under the CDDL license and is under the stewardship of maintainer Samara Kepler. The main dependency guiding its performance is ArborPilotStack. |
| | 1-hop Train Question | Q: What is the latest release of MapleSearchMesh? A: 1.3.0. |
| | 2-hop Train Question | Q: Which version is BriskX's main dependency currently at? A: AuroraEngineVault, the dependency of BriskX, is at version 4.4. |
| | 1-hop Test Question | Q: What programming language is PioneerKitKit written in? A: |

| | 2-hop Test Question | Q: Which language is PioneerKitKit's main dependency written in? A: To answer this question, first |
|---|---|---|

## A.3 STATISTICS

Across all four domains, each dataset is composed of 30,000 documents, generated from a pool of 3,000 entities with 10 document variations per entity. The total token count varies slightly by domain, ranging from 1.63M tokens in the *Software* dataset to 1.93M tokens in the *People* dataset, reflecting differences in attribute richness and text generation.

In addition to documents, each domain includes a large set of training questions. The number of training questions ranges from 65,000 in People to 85,000 in Planets, with token counts between 2.27M and 2.74M, covering both single-hop and multi-hop examples. For each attribute, we generate five QA pairs, distributed across single-hop and two-hop formulations. To evaluate generalization, we further reserved between 10,500 and 14,000 test questions, depending on the domain.

Overall, this setup provides a consistent scale across domains, while still allowing for diversity in textual content and reasoning challenges. The uniform number of documents facilitates direct cross-domain comparisons, whereas the variation in question amount and token count captures domain-specific complexity.

Table 6: Dataset statistics across domains.

| Domain | Documents | | Train Questions | | Test Questions |
|---|---|---|---|---|---|
| | # | tokens | # | tokens | # |
| People | 30,000 | 1,928,712 | 65,000 | 2,397,207 | 12,000 |
| Clubs | 30,000 | 1,721,156 | 67,500 | 2,389,915 | 10,500 |
| Planets | 30,000 | 1,732,157 | 85,000 | 2,748,070 | 14,000 |
| Software | 30,000 | 1,636,718 | 72,500 | 2,275,996 | 11,500 |

## A.4 GENERATION PROMPT

Our fictional data generation process begins with structured attributes, which we use as the foundation to create realistic documents. An external LLM is employed in this step and prompted to generate $N$ variations by adopting different writing styles and phrasings, thereby increasing data diversity. Early experiments indicated that such variation is essential for promoting generalization of the learned facts.

Below, we present the prompt used for document generation in the *People* domain. The other domains follow nearly identical instructions, differing only in the attributes provided and minor adjustments in wording to match their respective contexts.

```
Create {n} biographical paragraphs using the following personal
    information. The text does not need to follow the exact order
    provided, but make sure to mention all important information (name,
    birthday, workplace, birth state, parent's name, spouse's name).
    Generate {n} text variations with different phrasing and style,
    ranging from a simple and straightforward writing style to a more
    complex and detailed one. Vary the wording across different texts.
    Output the result in JSON format.

Name: {name}
Gender: {gender}
Works at: {company}
Birthdate: {birthdate}
Born in: {state}
Parent name: {parent_name}
```

```
Parent gender: {parent_gender}
Married to: {spouse_name}
Spouse gender: {spouse_gender}
```

## B  USE OF LARGE LANGUAGE MODELS

For this manuscript, we used LLMs as assistive tools to improve readability. In particular, LLMs were employed to polish the writing, correct grammar, and fix typos. They were not used for generating research ideas, designing experiments, or drafting substantive scientific content.

