# OpenReview forum: "Learning to Link: Incorporating Multi-hop QA Examples Improves Dispersed Knowledge Injection"
_ICLR.cc/2026/Conference — Submitted to ICLR 2026_

### Official Review · Reviewer_tXhJ · 2025-10-27

**Soundness:** 2
**Presentation:** 2
**Contribution:** 2
**Rating:** 4
**Confidence:** 4

**Summary:**

The paper studies how LLMs acquire the ability of multi-hop reasoning by connecting facts across different docs. Experiments on four synthetic datasets over different topics show that standard training on isolated document is insufficient. Incorporating questions with their grounding docs is essential during training. Additionally, the authors demonstrate that the connections cannot be effectively carried out in a single pass.

**Strengths:**

- The paper provides insights into the question: how do LLM learn to compose facts from different sources?
- The main finding that grounded QA examples are crucial for teaching this linking skill is significant
- Ablation provides strong evidence that grounding in documents is the key to better performance

**Weaknesses:**

- Experiments are carried out on purely synthetic data, which may have very different information layout as real-life documents.
- The paper adopts standard LM objective for training, yet models may further benefit from other training methods such as contrastive learning, RL, etc.
- All conclusions are drawn over a single model (Qwen2.5-7B), which may not generalize to models from different family or models of different sizes.

**Questions:**

- How does instruction model perform on multi-hop QA, as compared to base model?

---

> ### Author Response · Authors · 2025-11-21
>
> Thank you for reviewing our work and for the valuable feedback. Addressing the weaknesses and the question you raised:
> 1. The experiments were carried out on synthetic data aiming to isolate the new knowledge using completely unseen entities. By constructing fictional datasets, we ensured the model had not been exposed to these entities and relations before. This allowed us to securely measure the injection of new facts in a controlled setting. The training documents were constructed using an LLM to generate natural-looking text based on the structured attributes of each fictional entity, contextualized for each domain (e.g., in the People domain we prompted the LLM to generate biographies).
> 2. In this work, we adopted the standard continued pre-training strategy because it is the most common way to inject new knowledge, as seen in prior work on domain adaptation. We agree that exploring further training strategies combined with continued pre-training — such as RL or contrastive learning — is a promising direction for future work.
> 3. Even though we did not conduct experiments with different model families, we see no reason to believe a completely different behavior would emerge from switching architectures. For example, we do not expect a Llama or Olmo-based model to suddenly go from random performance in the multiple-choice setup to showing a significant leap absent in Qwen. We believe that changing the base model would mostly shift the absolute results rather than produce a qualitatively different pattern that would alter our conclusions. We agree that exploring different model sizes is important, but our current compute budget forces us to leave that for future investigation.
> 4. We used the instruct version of Qwen2.5-7B, which is capable of performing multi-hop QA on information it has seen during its training (as evidenced by its reported performance on HotpotQA and 2WikiMultiHopQA, for example). However, in our work we aimed to inject completely new facts that the model had no a priori knowledge of. We chose to start from the instruct variant because we wanted to test the requirement of chain-of-thought–style answering for solving multi-hop QA, as shown in Section 4.4.

---

### Official Review · Reviewer_msKC · 2025-10-31

**Soundness:** 3
**Presentation:** 3
**Contribution:** 2
**Rating:** 4
**Confidence:** 4

**Summary:**

The paper investigates how LMs learn to answer multi-hop questions with their parametric knowledge. The paper creates synthetic datasets on topics of “clubs”, “people”, “planet”, and “software”, and conducts controlled experiments on these datasets. The datasets are split into “documents” which describe the attributes of the entities, and “QA”s which are questions paired with answers and optionally how the intermediate (bridge) entity connects to the answer entity for two-hop questions. Results show that LMs cannot learn to link two entities in two different documents from QA formats alone. The QA pairs should be accompanied by the corresponding supporting documents. Analysis shows that LMs perform better in free-form generation settings compared to multiple-choice settings, since they cannot generate the intermediate entity.

**Strengths:**

1. The created dataset fits the purpose of the experiments. The dataset construction process is reasonable.
2. The descriptions of experiments are mostly clear, with some minor details left out. I don’t have to refer to the appendix a lot.
3. The experiment design fits the research questions asked, and the execution is pretty solid.

**Weaknesses:**

1. **Findings are limited.** This is my main complaint, and the main reason for my current score. I think RQ1 and RQ2 are pretty clear without doing the experiments, in that you would not expect the LM to learn cross-document links if it is only trained on individual documents. This is supported by previous works mentioned in L373-L377 in the paper. RQ3 seems obvious if you view it as in-domain training; if LMs only see the QA pairs without the documents, you are asking LMs to perform a task they have not seen before. LMs would not know at test time to connect different documents to answer the question, as they never did it during training. One evidence for supporting my view is that OOD generalization results are pretty bad, suggesting that whether the training is in-domain matters more than if you have both the documents and the QA pairs. RQ4 is the one I find more interesting, but I think multiple-choice questions are not the best way to test this and are not that realistic. There could be a better way to test if LMs could perform this in a single pass.
2. The experiments are only on one model. It is unclear how model architecture, pretraining data, instruction-following capabilities, and model size would affect the results. One example is that if the LM can perform chain-of-thought (CoT), it might not need many QA examples (fewer than number of documents perhaps) to answer multi-hop questions.
3. Results do not generalize OOD. To reiterate my point in 1., it seems whether the training is in-domain or not matters more than QA appearing alongside the documents. This is not necessarily a weakness, but this makes the claim that “pairing questions with their grounding documents is essential” weaker, and I think maybe the statements in the paper should be made weaker as well.

**Questions:**

1. What are the templates for generating the questions?
2. How are the attributes in Table 4 decided?
3. What are the prompts for generating the entities? (For clubs, planets, and software) Also how do you generate the family tree for people?
4. Do you have the statistics of how many other entities are each entity linked to? To get a sense of how many links are there between documents.
5. For docs-only results in Table 1, is the model trained on docs of training + test set?
6. To test if the model can answer the question in one step, should we also include QA pairs where the answers do not include the intermediate entity, but just the final answer? In that case, there is no train-test mismatch.
7.  In fine-tuning, how do you order the examples? How do you order QAs and documents?

---

> ### Author Response · Authors · 2025-11-21
>
> We appreciate the reviewer's careful reading and constructive feedback. Addressing the weaknesses and questions you raised:
> 1. Regarding RQ1 and RQ2 being obvious without experiments, we consider that our results and prior findings show that their answers are not as intuitive as they may seem. Table 1 answers RQ1 by showing that models can learn cross-document links when trained only on isolated documents, but only in a limited way — otherwise the two-hop results for documents-only training would be near zero. RQ2 concerns the generalization of including multi-hop QA supervision to other domains and tasks, which, to the best of our knowledge, is a novelty; this is answered in Section 4.2, where we evaluate OOD generalization.  Regarding RQ3 seeming obvious when viewed as purely in-domain training, we argue that this ablation is important because the QA examples used for training include a mixture of explicit chain-of-thought answers and direct answers (Appendix A.2). For example: “Q: In which state was Dennis Aiden Koch's spouse born? A: They are married to Shannon Clyde Pfeffer, who was born in Illinois.” This raises a meaningful question of whether the model is simply learning to answer with step-by-step reasoning or whether the QA examples genuinely help connect dispersed information across documents. Additionally, using QA examples to teach models new knowledge is not equivalent to “asking LMs to perform a task they have not seen before”; this approach has been previously explored in the literature (https://arxiv.org/abs/2401.08406 and https://arxiv.org/abs/2404.00213). Our results also show that QA examples do not help when combined with documents from other domains — see (2) vs. (4) in Table 2.
>
> 	Finally, we agree that multiple-choice questions may be an unrealistic way to test broad LM capabilities, but we consider them useful for probing whether the probability of the correct answer increases with training. In our experiment, we compared the output probability of the correct answer to four distractors. If the probability of the correct answer increased more than that of the incorrect ones due to training — i.e., if the model had learned to reach the correct answer latently — we would expect an improvement. However, results remain at random baseline levels in Table 3.
> 2. Expanding experiments to other models and scales is indeed an interesting direction for future work. However, we do not expect substantial differences when using models of similar scale — for example, we do not expect a shift from random multiple-choice performance to a significantly higher score solely by switching architectures at the same parameter count.
> 3. Regarding the lack of generalization to OOD, the results shown in Figure 2 and Table 2 actually indicate the opposite. Figure 2 shows that training on QA examples paired with grounding documents from three different domains (“Others”) improves performance in the target domain relative to training only on that domain's documents (Clubs: 43.0 -> 64.2; People: 25.0 -> 29.7; Planets: 64.9 -> 69.5; Software: 11.7 -> 35.2). These results show that, even though transfer from other domains does not reach in-domain performance, there are still gains from including QA+documents from different sources.
>
> Questions:
> 1. We will include these templates in an updated version of the manuscript.
> 2. The attributes for each domain are arbitrary and chosen based on the nature of the domain.
> 3. Entity creation is the most hand-crafted part of our pipeline. For the People domain, we used the FakerJS library. For the remaining domains, we used an auxiliary LLM with manual verification and editing. Family trees were created by sampling generations sequentially: for each of the 1,000 families, we generated three generations. The full set of entities and attributes will be made available in structured form in an updated version of the supplementary material.
> 4. In Table 4, we indicate the attributes used for linking. In the People domain, each entity has two connections (a parent and a spouse), and in the other domains each entity is directly linked to one other entity.
> 5. Yes, documents from the test set are always included. We will clarify this in Table 2 (line 2).
> 6. We did not include this ablation in the paper, but early experiments showed that it made no difference in the multiple-choice setup, while including more step-by-step style answers improved performance only in open-ended evaluation.
> 7. The ordering during training is random, documents and QA examples are shuffled and fed to the model.

---

### Official Review · Reviewer_QxbG · 2025-10-31

**Soundness:** 3
**Presentation:** 3
**Contribution:** 3
**Rating:** 6
**Confidence:** 3

**Summary:**

This paper tests the multi-hop QA ability of the large language model to verify its internal mechanism. The paper found that the generalization ability can be improved by letting the model contact the multi-hop QA samples and the corresponding documents, and proposed that it is not enough to provide only QA samples, and the questions must be paired with their basic documents during training.

**Strengths:**

1. This paper makes a detailed investigation on the mechanism of multi-hop QA ability.
2. This paper constructs a simulation dataset to verify the multi-hop capability.

**Weaknesses:**

1. The simulation dataset constructed in the paper did not comprehensively test its similarity with the real scene.
2. The paper explored single-hop and double-hop scenarios, but did not discuss further multi-hop situations.
3. Four research questions are put forward in the paper, but experiments are not conducted for the four questions in the experiment, and there is also a lack of analysis.

**Questions:**

Why choose these four entity concepts?  The concept of planets as a physical entity is not very common in real-life scenarios.

---

> ### Author Response · Authors · 2025-11-21
>
> Thank you for your thoughtful review and for highlighting the strengths of our work. Addressing the three weaknesses and the question you raised:
> 1. Similarity between our synthetic datasets and real text: We acknowledge that we did not run a dedicated evaluation of realism. However, Appendix A.2 includes examples of the generated documents, and Appendix A.4 presents the exact prompt we use to convert structured entity attributes into biography or news-style text. All documents are produced by an LLM conditioned on structured facts, which reliably yields natural-looking, human-like prose.
> 2. Limitation to two-hop reasoning: We agree this is a current limitation. Our goal was to isolate and characterize the learning dynamics that emerge when knowledge is dispersed across documents. Using two documents already surfaces the key behaviors we study (e.g., failure to bridge documents without explicit supervision), and extending to more hops is an important direction for future work.
> 3. Research questions and experimental coverage: Although we do not explicitly reference the RQs in the analysis, each one is directly addressed in the experimental section:
>     - RQ1 is answered in Table 1, which shows that models trained only on isolated documents achieve strong single-hop accuracy but struggle on two-hop ability.
>     - RQ2 is answered in Figure 2, which shows the gains obtained when QA supervision is added in-domain and out-of-domain.
>     - RQ3 is answered in Table 2, which provides a controlled ablation comparing QA-only, documents-only, and paired QA+documents, directly testing whether improvements come from the QA format itself or from grounding QA in documents.
>     - RQ4 is answered in Table 3, which compares multiple-choice (single-pass) evaluation with open-ended generation, showing how much multi-hop reasoning the model can perform in a single forward pass.
> 4. Choice of entity concepts: We selected four domains (people, clubs, software, planets) to provide diversity in relational structure while keeping the schema simple enough for controlled analysis. Planets, while not common in everyday interaction, offer clean, unambiguous physical relations useful for controlled multi-hop compositions. The goal was not to mirror real-world frequency but to create varied relational patterns for testing generalization.

---

### Official Review · Reviewer_Y6nD · 2025-11-04

**Soundness:** 3
**Presentation:** 2
**Contribution:** 2
**Rating:** 4
**Confidence:** 3

**Summary:**

This paper studies the question of how well LLMs can learn multi-hop facts that are spread across different documents. They show models are poor at this and propose a scheme to train models where they interleave documents and ask multi-hop questions. This improves performance on multi-hop questions substantially.

**Strengths:**

The paper could have insights into synthetic data generation schemes. Many open-source and frontier models are powered by synthetic data schemes that look to augment documents multiple times to better instill pieces of knowledge. This paper shows this needs to be done at a multi-document level rather than considering one document at a time.

**Weaknesses:**

* I think the presentation could be far improved. For example, I work closely in this area and didn't fully grok what the paper was talking about until reading the introduction.
* The experimental setup is very simplistic. We have a finetune of Qwen on a set of synthetic documents. One could imagine a synthetic data creation setting (e.g., something of flavor to this https://arxiv.org/abs/2508.09494 but not necessarily this exact work but just work in this flavor over the last few years), where we try to instill a set of facts at scale into a model via synthetic data.

**Questions:**

_

---

> ### Author Response · Authors · 2025-11-21
>
> Thank you for your thoughtful review and feedback. Addressing the two weaknesses you raised:
> 1. Regarding presentation: We appreciate the comment and agree that the clarity of our motivation and contributions can be improved. We are already revising the abstract and early sections to make the core idea — cross-document linking — more explicit from the beginning.
> 2. Regarding the simplicity of the experimental setup: Our goal was to isolate which pieces of knowledge a model can reliably acquire during training, rather than to evaluate performance at scale. To do this, we used controlled fictional datasets. We recognize the concern about generalization; in response, we created four different domains and evaluated cross-domain transfer to introduce variability and strengthen the evidence. Work in the flavor of the paper you referenced is indeed related, and we will incorporate that citation. However, scale is not the focus of our study: prior work has already shown that large-scale synthetic augmentation can successfully inject new knowledge. Our aim here is to examine the underlying mechanisms in a constrained setting, using entities the model certainly had not been exposed to during pre-training.

---

### Meta-Review · Area_Chair_oAHL · 2026-01-08

**Summary:**

The paper examines how language models (LMs) acquire the ability to answer multi-hop questions using their parametric knowledge. To explore this, the authors create synthetic datasets covering topics such as “clubs,” “people,” “planet,” and “software,” and carry out controlled experiments on these datasets. The datasets are divided into “documents,” which detail the attributes of various entities, and “QA” pairs, consisting of questions linked to their answers, and optionally, explanations of how the intermediate (bridge) entity relates to the answer entity in two-hop questions. The findings reveal that LMs struggle to connect two entities across different documents when relying solely on QA formats. It is essential for QA pairs to be supplemented with their corresponding supporting documents. Furthermore, the analysis indicates that LMs perform more effectively in free-form generation scenarios than in multiple-choice formats, as they are unable to generate the intermediate entity.

**Reviewer Concerns:**

The reviewers believe the experimental setup is very simplistic.
The simulated dataset is far from real worlds.
The paper can be extended to multi-hop situations.

**Reviewer Scores:**

The scores of the reviewers are 4,6,4,4, and in the reviewers do not change their scores.

---

### Decision · Program_Chairs · 2026-01-26

Reject